# Serotonin modulates insect hemocyte phagocytosis via two different serotonin receptors

Yi-xiang Qi[†], Jia Huang*[†], Meng-qi Li, Ya-su Wu, Ren-ying Xia, Gong-yin Ye*

State Key Laboratory of Rice Biology, Institute of Insect Sciences, Zhejiang University, Hangzhou, China

**Abstract** Serotonin (5-HT) modulates both neural and immune responses in vertebrates, but its role in insect immunity remains uncertain. We report that hemocytes in the caterpillar, *Pieris rapae* are able to synthesize 5-HT following activation by lipopolysaccharide. The inhibition of a serotonin-generating enzyme with either pharmacological blockade or RNAi knock-down impaired hemocyte phagocytosis. Biochemical and functional experiments showed that naive hemocytes primarily express $5\text{-HT}_{1B}$ and $5\text{-HT}_{2B}$ receptors. The blockade of $5\text{-HT}_{1B}$ significantly reduced phagocytic ability; however, the blockade of $5\text{-HT}_{2B}$ increased hemocyte phagocytosis. The $5\text{-HT}_{1B}$-null *Drosophila melanogaster* mutants showed higher mortality than controls when infected with bacteria, due to their decreased phagocytotic ability. Flies expressing $5\text{-HT}_{1B}$ or $5\text{-HT}_{2B}$ RNAi in hemocytes also showed similar sensitivity to infection. Combined, these data demonstrate that 5-HT mediates hemocyte phagocytosis through $5\text{-HT}_{1B}$ and $5\text{-HT}_{2B}$ receptors and serotonergic signaling performs critical modulatory functions in immune systems of animals separated by 500 million years of evolution.

*For correspondence: huangj@zju.edu.cn (JH); chu@zju.edu.cn (GyY)

[†]These authors contributed equally to this work

Competing interests: The authors declare that no competing interests exist.

## Introduction

Serotonin (5-hydroxytryptamine, 5-HT) is one of the oldest neurotransmitters/hormones in evolution (*Turlejski, 1996*). It regulates or modulates a wide variety of processes in most invertebrates and vertebrates, such as metabolism (*Sze et al., 2000*) and locomotion (*Ranganathan et al., 2000*) in nematodes; reproduction (*Anstey et al., 2009*), learning and memory (*Sitaraman et al., 2008*) in insects; physiologic states and behaviors, including pain, appetite, mood, and sleep (*Mössner and Lesch, 1998*) in humans. 5-HT also plays an important role outside of the central nervous system (CNS) in immune signaling. Immune cells can synthesize and sequester 5-HT. For instance, human mast cells express the key peripheral 5-HT synthesizing enzyme, tryptophan hydroxylase 1 (TPH-1) (*Kushnir-Sukhov et al., 2007*, *2008*). Mouse dendritic cells (DCs) express the serotonin transporter (SERT), taking up 5-HT from the microenvironment (*O'Connell et al., 2006*). 5-HT regulates immune responses and inflammatory cascades via distinct receptors and different immune cells have been shown to express a different composition of 5-HT receptor subtypes (*Baganz and Blakely, 2013*). $5\text{-HT}_{2A}$ may contribute to chemotaxis of eosinophils (*Boehme et al., 2008*) and $5\text{-HT}_{2C}$ receptors on alveolar macrophages can be activated by 5-HT (*Mikulski et al., 2010*). However, most studies assess the in vitro response of immune cells to pharmacological agents. Therefore, the function of 5-HT signaling in vivo in are still unclear.

Vertebrate blood cells (e.g. macrophages) have evolved a variety of strategies to internalize particles and solutes, including pinocytosis, receptor-mediated endocytosis, and phagocytosis, a highly conserved aspect of innate immunity. Phagocytosis, the uptake of large particles (>0.5 µm) into cells, allows for rapid engulfment of dying cells and pathogens by specialized phagocytes, such as

**eLife digest** Serotonin is a small molecule found in organisms across the animal kingdom. This molecule plays various roles in the human body and affects many systems including the gut and central nervous system. Over recent decades, serotonin has been found to play a role in the immune system too, and appears to help regulate how immune cells respond to invasion by infectious bacteria or viruses. Various types of immune cells that can engulf foreign particles or microorganisms via a process called phagocytosis have receptors for serotonin on their cell surface and are activated when serotonin is present.

Signaling pathways associated with part of the immune system in mammals are often highly similar to pathways found in insects. Serotonin is also known to influence many processes in insects, such as appetite, sleep and reproduction, but its role in the insect's immune system was not well understood. In particular, insects have phagocytic cells known as hemocytes and it was unknown if serotonin helps to activate these cells.

Qi, Huang et al. have now discovered that serotonin does indeed control the activity of insect hemocytes from the caterpillars of the small white butterfly *(Pieris rapae)* and the fruit fly *(Drosophila melanogaster)*. The experiments showed that two distinct receptors on a hemocyte's cell surface can detect serotonin. One of these receptors increases phagocytic activity in both insects, while the other has the opposite effect in the caterpillar and reduces this activity.Qi, Huang et al. also discovered that phagocytosis depends on which of these receptors is most common on the hemocyte cell surface, and demonstrated that insects exposed to bacteria start to produce more of the serotonin receptors that increase phagocytosis. Further experiments showed that fruit flies in which the gene for a serotonin receptor has been deleted are more vulnerable to bacterial infections due to their poor phagocytic ability.

Insects and mammals are separated by about 500 million years of evolution, and so these findings suggest that serotonin is an ancient signaling molecule that can control the immune system across the animal kingdom. The work also supports the idea that studies of the simpler immune systems of insects, including the model organisms such as *D. melanogaster*, can offer insight into the immune systems of humans and other animals.

macrophages and neutrophils in mammals (*Aderem and Underhill, 1999*). In insects, hemocyte phagocytosis is an important cellular defense response to pathogens and parasites (*Lavine and Strand, 2002*). In lepidopteran insects, the granulocytes and plasmatocytes are the major phagocytes (*Kanost et al., 2004*). Cross-talk between the immune and nervous system may play a role in regulating phagocytosis in insects during infection. However, the roles of serotonin in insect phagocytosis are less well characterized compared with vertebrate counterparts, although there is evidence that 5-HT can enhance phagocytosis (*Baines et al., 1992*; *Kim et al.,2009*).

Many of the intracellular signaling pathways that drive the insect immune system are very similar to those found in the mammalian innate immune system (*Lemaitre and Hoffmann, 2007*), and some of them were first uncovered in insects (*Hoffmann, 2003*). Indirect evidence suggests that similarities between insects and mammals extend to the molecular mechanisms involved in the neuroendocrine control of immune function (*Adamo, 2008*). Because of the relative simplicity of the insect immune system, examining the basic interactions between serotonin receptor-medicated second messenger systems and immune-related intracellular signaling pathways may be easier in insects. As an initial step toward this goal, we have made a comprehensive study in the caterpillar, *Pieris rapae* hemocytes to elucidate the function of 5-HT signaling in insect cellular immune responses. We found that hemocyte-derived 5-HT regulates phagocytosis in an autocrine manner through 5-HT$_{1B}$ and 5-HT$_{2B}$ receptors, each of which produces distinct effects. Mortality experiments using *Drosophila* mutants and hemocyte-specific RNAi-silencing further found that both 5-HT$_{1B}$ and 5-HT$_{2B}$ are necessary for effective resistance to bacterial infections.

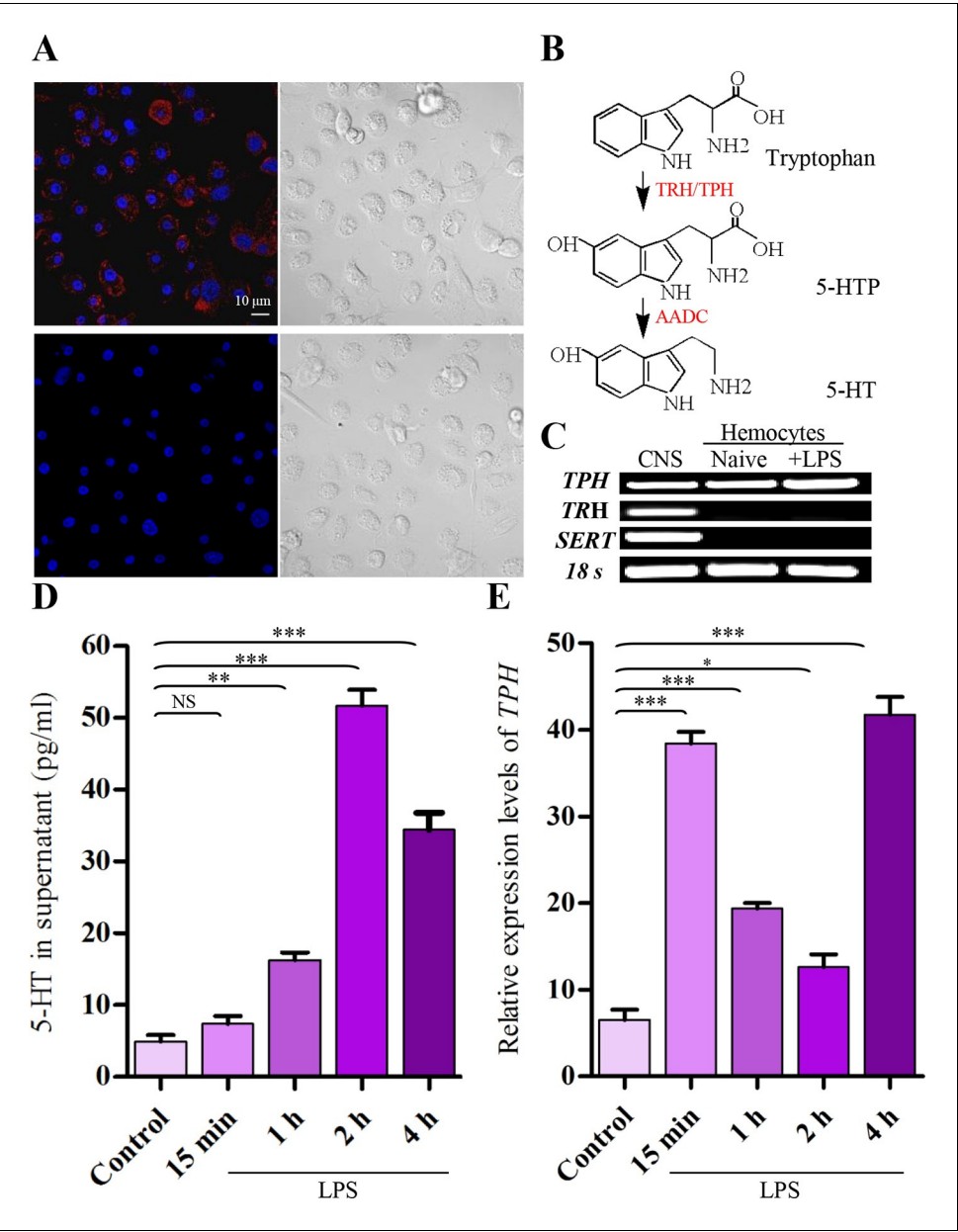

**Figure 1.** Activated hemocytes are capable of 5-HT synthesis. (**A**) 5-HT was visualized by confocal microscopy in hemocytes activated with 100 ng/ml LPS by labeling with 5-HT antisera (Alexa Fluor 546; red) (upper left). 5-HT antisera was preabsorbed with 5-HT as the negative control (lower left). Nuclei were counterstained with DAPI (blue). Scale bar represents 10 μM. Data are *representatives* of two independent experiments. (**B**) Serotonin biosynthetic pathway. Tryptophan- phenylalanine hydroxylase (TRH/TPH), aromatic L-amino acid decarboxylase (AADC), 5-hydroxy tryptophan (5-HTP). (**C**) Expression of gene transcripts for *TPH, TRH*, and *SERT* were determined by RT-PCR from naive hemocytes and from hemocytes activated with 100 ng/ml LPS for 2 hr, central nervous system (CNS) as positive control. Data are *representatives* of three independent experiments. (**D**) 5-HT concentrations in hemocytes supernatants were determined by ELISA. Hemocytes were activated with 100 ng/ml LPS. Naive hemocytes treated with PBS are as control (*n* = 4). (**E**) Relative expression of TPH was quantified by real-time PCR. Hemocytes were activated with 100 ng/ml LPS. Naive hemocytes treated with PBS is as control (*n* = 3). One-way ANOVA followed by Tukey's multiple comparison test for **D** and **E**. Error bars indicate ± s.e.m., ***p<0.001, **p<0.01, *p<0.05 and NS means no significant difference.

## Results

### Hemocytes synthesize 5-HT in an activation-dependent manner

After activation with 100 ng/ml of lipopolysaccharide (LPS) for 2 hr, we detected 5-HT within hemocytes directly by immunolabeling and fluorescence microscopy. *Figure 1A* shows that granules of 5-HT labeled with 5-HT antisera (upper) are readily visible in the cytosol. As a negative control, 5-HT antisera preabsorbed with 5-HT were invisible (lower). Serotonin synthesis requires two enzymes,

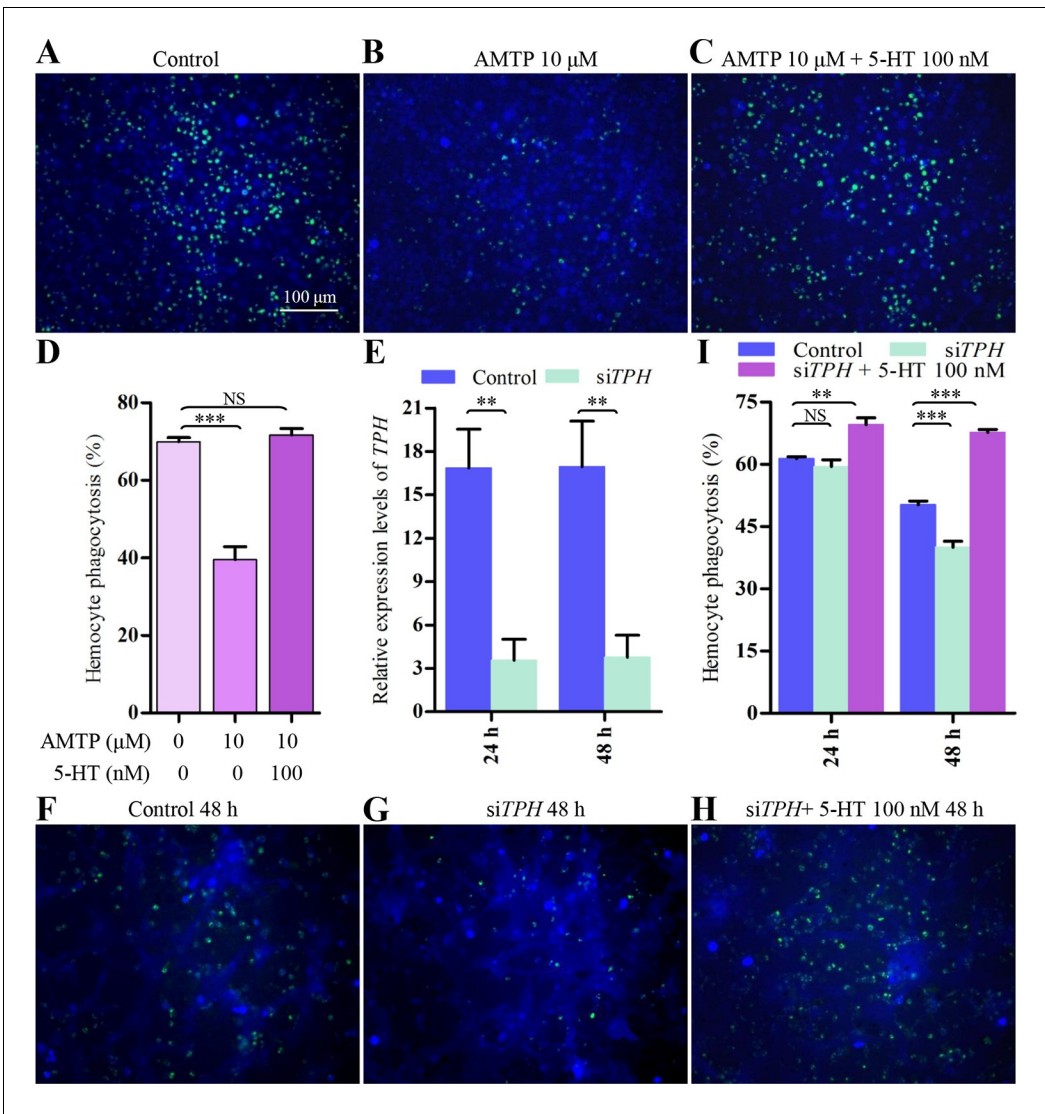

**Figure 2.** Inhibition of endogenous 5-HT synthesis impairs hemocyte phagocytosis. (A-C) Hemocyte phagocytosis was visualized by florescence microscope. Hemocytes were stained with Cell Tracker Blue CMAC (blue), green represent the phagocytosed pHrodo *E. coli*. (D) Quantification of phagocytosis of *E. coli* by hemocytes (*n* = 3). (E) Confirmation of knock-down effect of *TPH* by real-time qPCR. The *P. rapae18s rRNA* gene was used as an internal reference gene (*n* = 4). (F-H) Effect of si*TPH* on hemocyte phagocytosis was visualized by florescence microscope. (I) Quantification of si*TPH* effect on hemocyte phagocytosis (*n* = 3). One-way ANOVA followed by Tukey's multiple comparison test for D and I; two-tailed *t*-test for E. Error bars indicate ± s.e.m., ***p<0.001, **p<0.01 and NS means no significant difference.

The following figure supplement is available for figure 2:

**Figure supplement 1.** Inhibition of 5-HT synthesis by AMTP.

tryptophan hydroxylase and aromatic L-amino acid decarboxylase (AADC) (*Figure 1B*). Tryptophan hydroxylase is the rate-limiting enzyme in serotonin biosynthesis encoded by two genes, TPH and TRH in insects, TPH1 and TPH2 in mammals. Both gene products have tryptophan hydroxylase activity in vivo. The TPH/TPH1 gene is expressed in non-neural tissues, and the TRH/TPH2 gene is expressed in neural tissues (*Côté et al., 2003*; *Neckameyer et al., 2007*; *Watanabe et al., 2011*). The high-affinity SERT is a plasma membrane protein that can take up extracellular 5-HT (*Rudnick, 2006*; *Torres et al., 2003*). Thus, we performed RT-PCR to characterize TPH, TRH, and SERT expression in hemocytes. *Figure 1C* shows that hemocytes produced mRNA for *TPH*, but that transcripts of *TRH* and *SERT* were not detected. These results suggest that hemocytes do not selectively sequester 5-HT but can synthesize 5-HT using TPH. Then, we performed ELISA to quantify the amount of 5-HT released into the hemocytes' culture media. 5-HT levels increased significantly above that found in controls in cell supernatants 1 hr after exposure of hemocytes to LPS. 5-HT levels reached maximal at 2 hr and decreased at 4 hr (*Figure 1D*). The real-time quantitative RT-PCR results also show that mRNA expression level of *TPH* was up-regulated by approximately sixfold relative to controls at 15 min and at 4 hr after LPS stimulation, suggesting a potential negative feedback regulation (*Figure 1E*). Therefore, it is concluded that hemocytes are able to synthesize and release 5-HT in vitro, and this activity is enhanced following their activation. We hypothesise that 5-HT may play an important role in hemocyte function.

## 5-HT is involved in hemocyte phagocytosis

To examine the physiological role of 5-HT in hemocytes, we tested whether inhibition of 5-HT synthesis would affect hemocyte phagocytosis of Gram-negative *E. coli* bacteria labeled with pHrodo, a dye that fluoresces in the acidic environment of a mature phagosome upon fusion with lysosomes. α-methyltryptophan (AMTP) is a competitive inhibitor of 5-HT synthesis (*Gal and Christiansen, 1975*). As shown in *Figure 2—figure supplement 1*, 5-HT synthesis was significantly reduced in hemocytes after treatment with 10 μM AMTP, compared with PBS-treated controls. The results show that 10 μM AMTP also significantly impaired hemocyte phagocytic ability. Furthermore, treatment with exogenous 5-HT at 100 nM fully restored hemocyte phagocytosis (*Figure 2A–D*). Next, we tested si*TPH*-treated hemocytes (knock-down effect was approximately 80%, *Figure 2E*) for their phagocytosis ability. After incubation with si*TPH* for 48 hr, hemocyte phagocytosis rate was significantly decreased compared with the negative control. 100 nM 5-HT treatment could fully rescue the si*TPH* induced phenotype (*Figure 2F–I*). Collectively, these data indicate that hemocyte-derived 5-HT is critical for proper hemocyte phagocytosis upon immune challenge.

## 5-HT regulates hemocyte phagocytosis via Pr5-HT$_{1B}$ and Pr5-HT$_{2B}$

Since 5-HT is involved in the regulation of hemocyte function, it should exert its effects through corresponding 5-HT receptors. So far, five subtypes of 5-HT receptors including 5-HT$_{1A}$, 5-HT$_{1B}$, 5-HT$_{2A}$, 5-HT$_{2B}$ and 5-HT$_7$ are identified in insects (*Blenau and Thamm, 2011*; *Gasque et al., 2013*). 5-HT$_{1A}$ and 5-HT$_{1B}$ couple with G$_i$ protein and decrease intracellular cAMP levels. 5-HT$_{2A}$ and 5-HT$_{2B}$ couple with G$_q$ protein, which can induce increased intracellular Ca$^{2+}$. 5-HT$_7$ couples with G$_s$ protein and increases cAMP levels (*Blenau and Thamm, 2011*; *Gasque et al., 2013*). We performed a comprehensive analysis of 5-HT receptor gene expression in hemocytes by RT-PCR after stimulation with 100 ng/ml LPS for 2 hr. The positive control samples were extracted from the CNS of *P. rapae*. As shown in *Figure 3A*, naive hemocytes express *Pr5-HT$_{1B}$*, *Pr5-HT$_{2B}$* and *Pr5-HT$_7$*. We used specific antibodies to confirm the expression of Pr5-HT$_{1B}$ (*Figure 3B*) and Pr5-HT$_{2B}$ (*Figure 3C*) on the plasma membranes of hemocytes. We also performed Ca$^{2+}$ imaging to further confirm functional expression of 5-HT$_{2B}$ in hemocytes. Both plasmatocytes and granulocytes, the two most abundant types of hemocytes, produced intracellular Ca$^{2+}$ increase in response to 5-HT. 5-HT stimulated Ca$^{2+}$ responses at concentrations ranging from 0.01 nM to 10 nM (maximal response), and its EC$_{50}$ value was estimated at 0.15 nM (*Figure 3—figure supplement 1*).

Since there are multiple 5-HT receptor types on the hemocyte membranes, we next investigated which one was involved in the 5-HT-mediated phagocytosis by pharmacological manipulation. Three selective 5-HT receptor antagonists (SB 216641, SB 269970, and RS 127445) were applied at a final concentration of 10 μM to block their corresponding receptors. The inhibition effects of SB 216641 and SB 269970 on Pr5-HT$_{1B}$ and Pr5-HT$_7$ were also confirmed by cAMP assays in heterogeneous

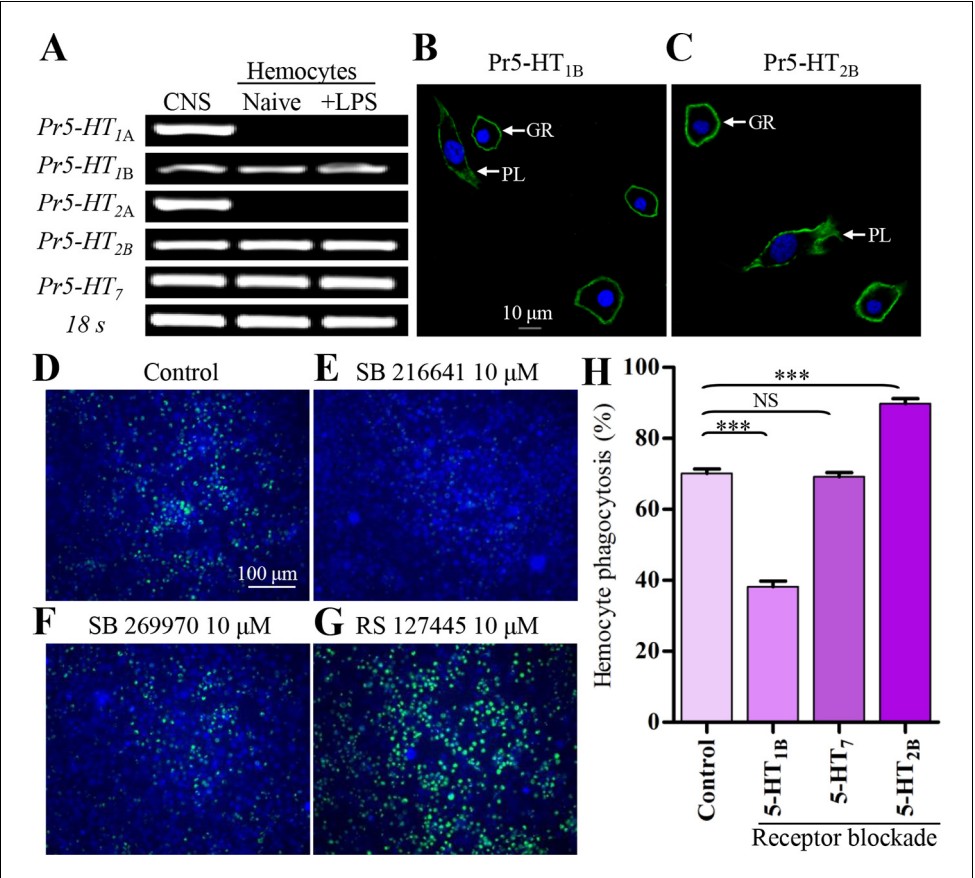

**Figure 3.** 5-HT receptor subtypes expressed in naïve and LPS-activated hemocytes. (**A**) Hemocytes were negatively purified and activated with 100 ng/ml LPS for 2 hr. The gene expression for 5-HTR subtype was examined by RT-PCR. Data are *representatives* of three independent experiments. (**B**) Gene expression for *Pr5-HT_1B* was examined by immunofluorescence. The scale bar represents 10 μM. Data are *representatives* of two independent experiments. (**C**) Gene expression for *Pr5-HT_2B* was examined by immunofluorescence. PL, plasmatocytes; GR, granulocytes. Data are *representatives* of two independent experiments. (**D-G**) The effect of different antagonist on hemocyte phagocytosis was visualized by florescence microscope. SB216641 is an antagonist of 5-HT_1B. SB269970 is an antagonist of 5-HT_7 and RS127445 is a human 5-HT_2B antagonist. (**H**) Quantification of different antagonist on hemocyte phagocytosis. Data are from three independent experiments that each consists of cells from ten fifth-instar larvae. One-way ANOVA followed by Tukey's multiple comparison test for **H**. Error bars indicate ± s.e.m., ***$p<0.001$, **$p<0.01$ and NS means no significant difference.

The following figure supplements are available for figure 3:

**Figure supplement 1.** Representative $Ca^{2+}$ responses and dose-response profiles for 5-HT in hemocytes.

**Figure supplement 2.** Modulation of intracellular cAMP levels in HEK 293 cells stably expressing the Pr5-HT_1B and Pr5-HT_2B.

expression system (*Figure 3—figure supplement 2*). We found that only blockade of Pr5-HT_1B significantly reduced the phagocytic ability of hemocytes. Blockade of Pr5-HT_7 neither intensified nor reduced hemocyte phagocytosis. Surprisingly, blockade of 5-HT_2B significantly increased hemocyte phagocytosis (*Figure 3D–H*).

We further found that the 5-HT_1B blocker SB 216641 affected hemocyte phagocytosis in a dose-dependent manner (*Figure 4A*). To confirm the above results, we performed siRNA-mediated interference to knock-down each receptor in hemocytes. The results showed that si*Pr5-HT_1B* decreased Pr5-HT_1B expression significantly at both mRNA and protein levels (*Figures 4B–C*) after 24 hr and 48 hr, respectively. As expected, the si*Pr5-HT_1B* treated hemocytes phagocytose *E. coli* poorly

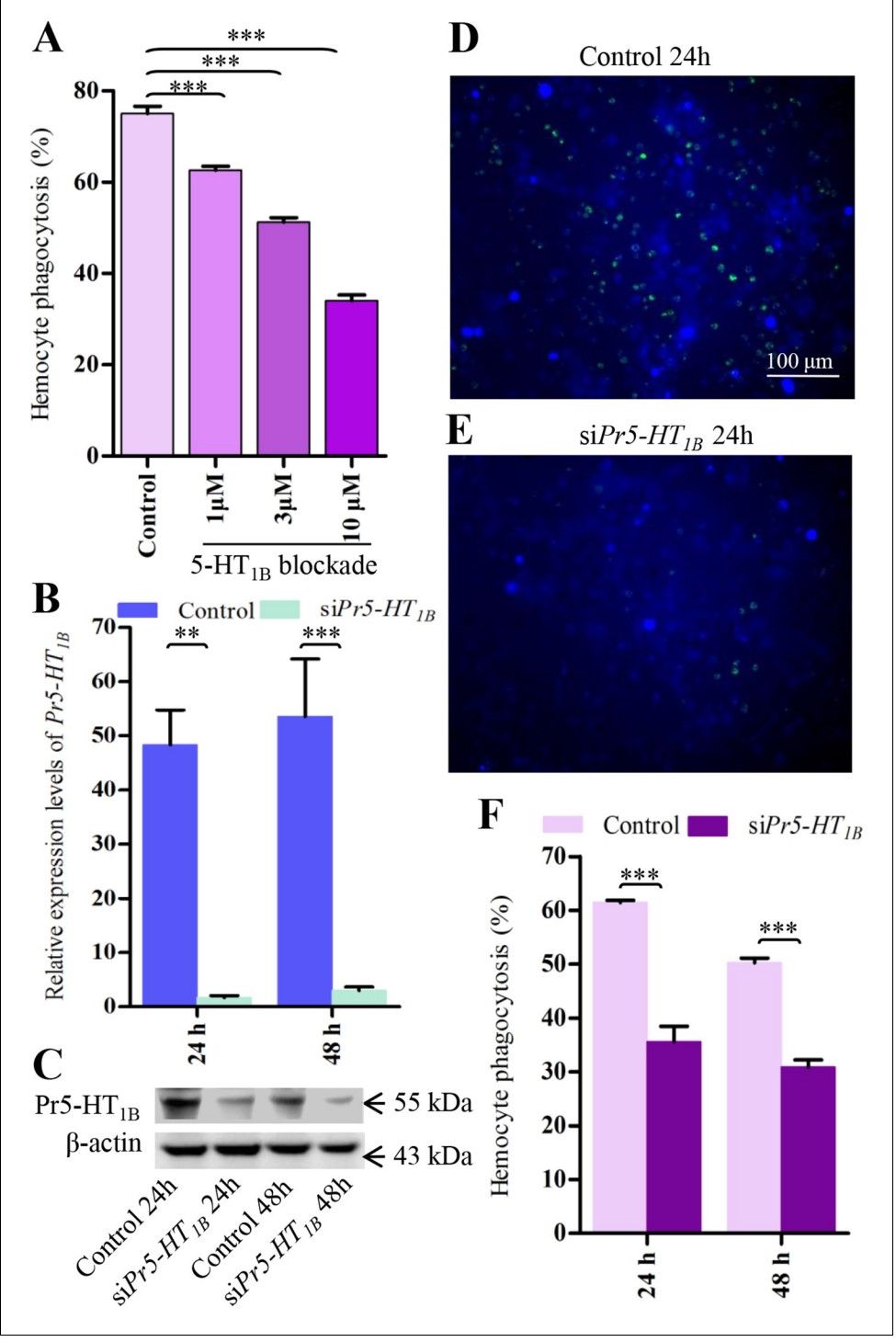

**Figure 4.** Pr5-HT$_{1B}$ mediates hemocyte phagocytosis. (A) Dose-response profiles for the effects of Pr5-HT$_{1B}$ blocker SB 216641 on hemocyte phagocytosis. Data are from three independent experiments that each consists of cells from ten fifth-instar larvae. (B) Confirmation of knock-down effect of *Pr5-HT$_{1B}$* by real-time PCR. The *P. rapae* 18s rRNA gene was used as an internal reference gene (*n* = 4). (C) Western blot analysis of knock-down effect of Pr5-HT$_{1B}$. β-actin was used to show equal protein loading. (D-E) Effect of si*Pr5-HT$_{1B}$* on hemocyte phagocytosis was visualized by florescence microscope. (F) Quantification of si*Pr5-HT$_{1B}$* effect on hemocyte phagocytosis. Data are from three independent experiments that each consists of cells from ten fifth-instar larvae. One-way ANOVA followed by Tukey's multiple comparison test for **A**; two-tailed *t*-test for **B** and **F**. Error bars indicate ± s.e.m., \*\*\*p<0.001, \*\*p<0.01.

*Figure 4 continued on next page*

*Figure 4 continued*

The following figure supplement is available for figure 4:

**Figure supplement 1.** Effect of si*Pr5-HT₂ₐ* and si*Pr5-HT₇* on hemocyte phagocytosis.

compared with control (*Figure 4D–F*). Significantly knock-down of *Pr5-HT₂ₐ* was also observed at both transcript and protein levels (*Figure 4—figure supplement 1A–B*) at 48 hr, which promoted hemocyte phagocytosis (*Figure 4—figure supplement 1C–E*). However, knock-down of *Pr5-HT₇* have no significant effect on hemocyte phagocytosis (*Figure 4—figure supplement 1F–G*). The results demonstrate that 5-HT mediates hemocyte phagocytosis through 5-HT₁ₐ and 5-HT₂ₐ receptors but that these two receptors act in opposite ways.

## Effect of immune challenge on Pr5-HT₁ₐ and Pr5-HT₂ₐ expression

Although the above data showed that activation of Pr5-HT₁ₐ and Pr5-HT₂ₐ produced opposite effects, enhanced hemocyte phagocytosis is the overall effect by hemocytes exposed to released 5-HT. Thus, we tested whether 5-HT receptors were upregulated or downregulated in hemocytes during an immune response. qPCR results showed that the mRNA levels of *Pr5-HT₁ₐ* was significantly

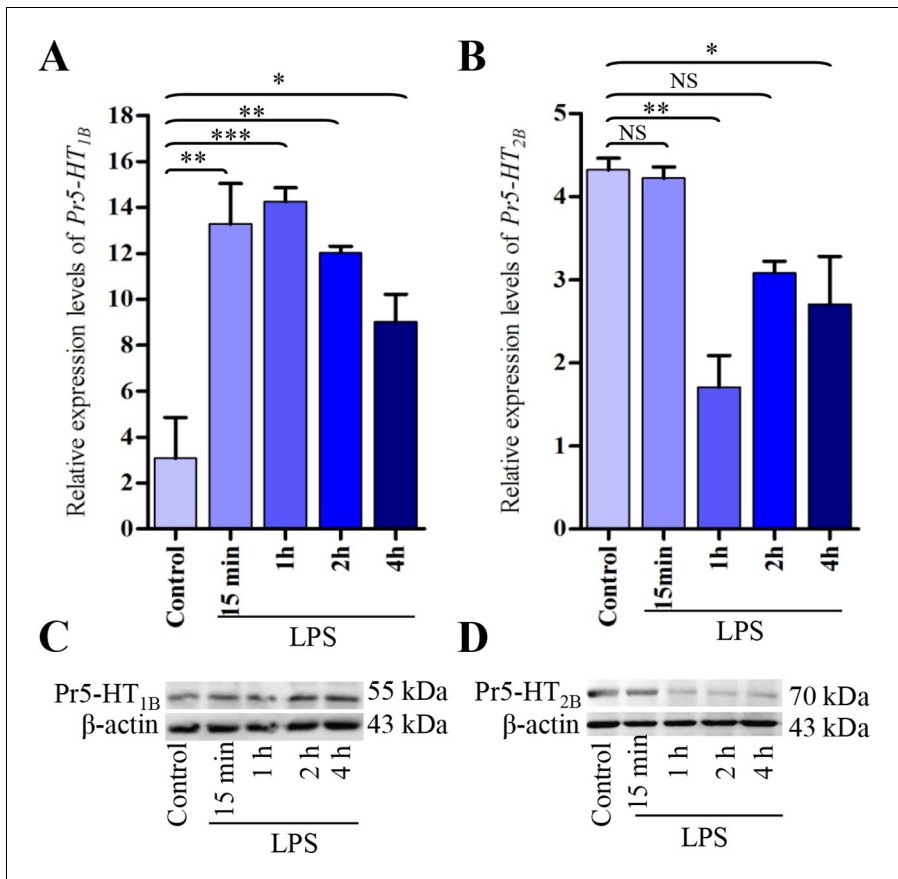

**Figure 5.** Expression analysis of Pr5-HT₁ₐ and Pr5-HT₂ₐ in naïve and LPS-induced hemocytes. (**A–B**) Relative expression of *Pr5-HT₁ₐ* and *Pr5-HT₂ₐ* were quantified by q-PCR. The *P. rapae* 18s rRNA gene was used as an internal reference (*n* = 3). (**C–D**) Western blot analysis of Pr5-HT₁ₐ and Pr5-HT₂ₐ in naive and LPS-induced hemocytes. β-actin was used to show equal protein loading. One-way ANOVA followed by Tukey's multiple comparison test for **A** and **B**. Error bars indicate ± s.e.m., ***p<0.001, **p<0.01, *p<0.05, and NS means no significant difference.

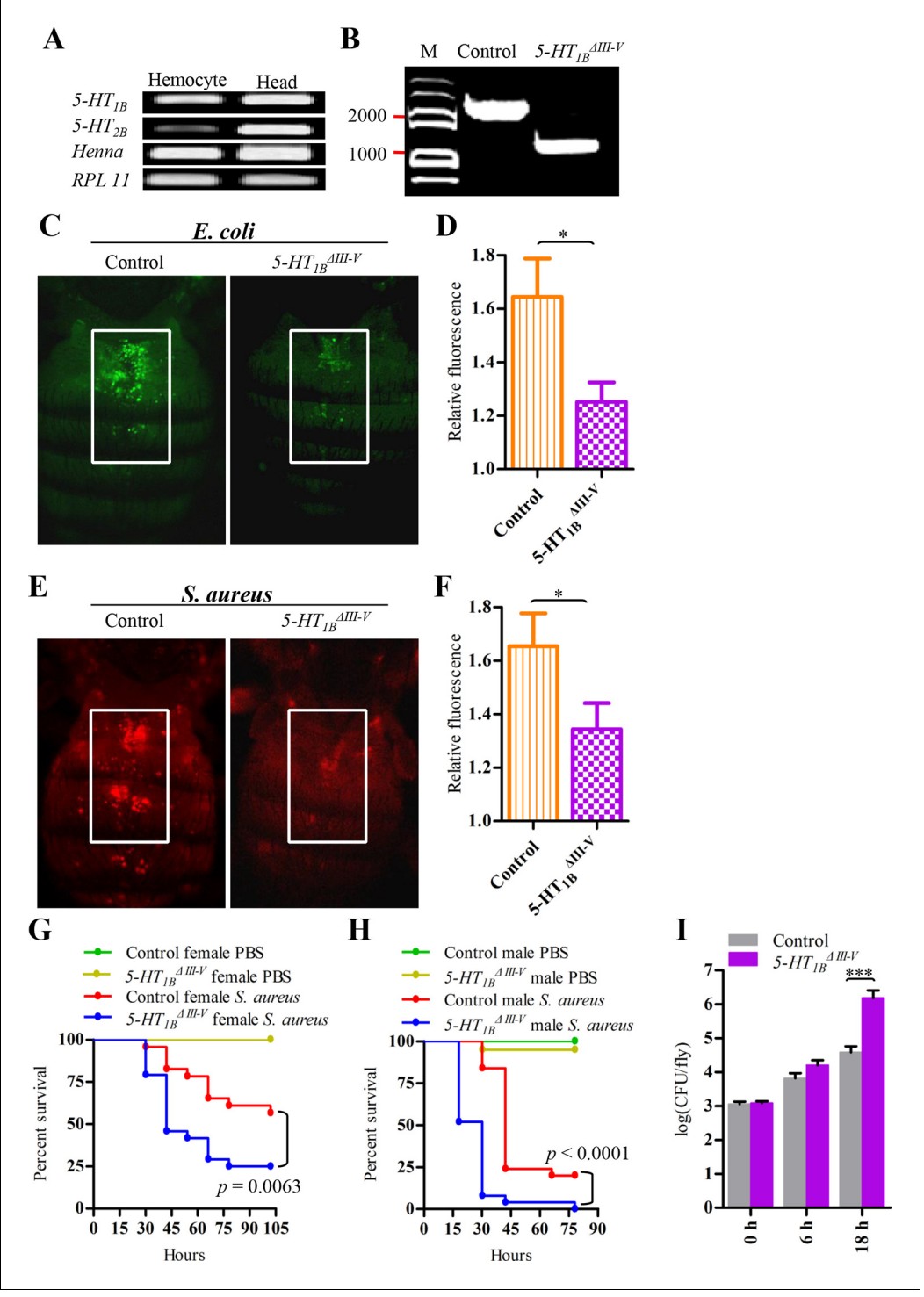

**Figure 6.** 5-HT$_{1B}$ is required for microbial phagocytosis and plays an important role in the *Drosophila* defense against *S. aureus* infection. (**A**) 5-HT$_{1B}$, 5-HT$_{2B}$ and *TPH* are expressed in *Drosophila* naive hemocytes. *RPL11* was used as an internal reference gene. Data are *representatives* of three independent experiments. (**B**) Genomic PCR of 5-HT$_{1B}$ control and 5-HT$_{1B}^{\Delta III-V}$ flies. Data are *representatives* of three independent experiments. (**C**) Representative pictures depicting phagocytosis in 5-HT$_{1B}$ control and 5-HT$_{1B}^{\Delta III-V}$ flies of fluorescein-labeled *E. coli* bioparticles. (**D**) Quantification of in vivo phagocytosis of *E. coli*. Approximately 10 flies per genotype were used in each experiment. Data are *representatives* of three independent experiments. (**E**) Representative pictures depicting phagocytosis in 5-HT$_{1B}$ control and 5-HT$_{1B}^{\Delta III-V}$ flies of fluorescein-labeled *S. aureus* bioparticles. (**F**) Quantification of in vivo phagocytosis of *S. aureus*. Approximately eight flies per genotype were used in each

*Figure 6 continued on next page*

*Figure 6 continued*

experiment. Data are *representatives* of three independent experiments. (G-H) Representative survival curves of female (G) and male (H) 5-HT$_{1B}$ control and 5-HT$_{1B}^{\Delta III-V}$ flies after injection of *S. aureus* (optical density [OD] 0.4). $n$ = 20–25 flies. Experiments were performed in triplicate. (I) Comparison of the *S. aureus* (OD 0.4) recovered in 5-HT$_{1B}$ control and 5-HT$_{1B}^{\Delta III-V}$ flies 0, 6, and 18 hr post infection. Bacterial load was measured in eight individual female flies per genotype at each time point in each experiment. Two-tailed *t*-test for D, F and I. Error bars indicate ± s.e.m., \*\*\*p<0.001, \*\*p<0.01, \*p<0.05.

increased after LPS stimulation from 15 min to 4 hr (*Figure 5A*). The protein expression levels also increased at 2 hr and 4 hr (*Figure 5C*). Interestingly, the mRNA expression level of *Pr5-HT$_{2B}$* was significantly decreased after LPS induction at 1 hr and 4 hr (*Figure 5B*), in accordance with its protein expression levels (*Figure 5D*).

## 5-HT$_{1B}$ and 5-HT$_{2B}$ are required for proper bacteria phagocytosis and resistance in *Drosophila*

We hypothesized that 5-HT receptors mediate immunoregulation in other insects and that it is important in host defense in vivo. On the other hand, it is a technical challenge to perform in vivo RNAi in Lepidoptera (*Terenius et al., 2011*). Thus, we chose the model animal *Drosophila melanogaster* to address the above questions. Both *5-HT$_{1B}$* and *5-HT$_{2B}$* transcripts were detected in wild-type *Drosophila* hemocytes by RT-PCR. High mRNA levels of *Henna*, the *Drosophila TPH* homolog, were also expressed in hemocytes (*Figure 6A*). We next used a *5-HT$_{1B}$* null allele (*5-HT$_{1B}^{\Delta III-V}$*) and its corresponding control allele (*Gasque et al., 2013*) to examine the ability to phagocytose bacteria as well as susceptibility to infections. A 1344 bp fragment is removed from genomic DNA to disrupt the III-V transmembrane domains of the *5-HT$_{1B}^{\Delta III-V}$* allele (*Figure 6B*).

The phagocytic capacity of flies was measured using an in vivo adult phagocytosis assay (*Kocks et al., 2005*). Flies were injected with fluorescently labeled pHrodo bioparticles. The amount of fluorescence in the dorsal vessel area where sessile phagocytes accumulates were visualized and quantified. In *5-HT$_{1B}^{\Delta III-V}$* flies, in vivo phagocytosis of *E. coli* was strongly impaired when compared to controls (*Figure 6C–D*). A similar effect was observed with the Gram-positive bacterium *S. aureus* (*Figure 6E–F*). After infection with *S. aureus*, *5-HT$_{1B}^{\Delta III-V}$* flies die more quickly than controls, for both male and female flies (*Figure 6G–H*). To determine whether the increased mortality of *5-HT$_{1B}^{\Delta III-V}$* flies following infection was due to defective resistance or decreased tolerance (*Schneider and Ayres, 2008*), we also assessed bacterial clearance by comparing colony-forming units (CFU) 6 hr and 18 hr after infection. There is an increased bacterial load in *5-HT$_{1B}$* null flies compared to control files (*Figure 6I*).

5-HT$_{1B}$ function in the regulation of hemocyte phagocytosis was further confirmed by RNAi. We used the *HmlΔ-Gal4* (*Sinenko et al., 2004*) driver to specifically express *UAS-1B RNAi* (*Yuan et al. 2005*) and UAS-*5-HT1B RNAi25833* respectively, both of which led to decreased phagocytosis of *E. coli* and *S. aureus* (*Figure 7A–D*). To test whether the hemocyte-specific knockdown of *5-HT$_{1B}$* dampened phagocytosis in general, or whether there was a lack of inducibility after immune challenge, we injected flies with PBS or 5-HT and then latex beads. When flies were injected with PBS, all flies showed similar phagocytic capacity. However, when flies were injected with 5-HT, knocking down *5-HT$_{1B}$* failed to enhance latex beads phagocytosis as controls (*Figure 7E*). After injection with *S. aureus*, the flies expressing *5-HT$_{1B}$* RNAi in their blood cells were much weaker than the control flies (*Figure 7F–I*). Our RNAi data indicate that the increased susceptibility to bacteria in *5-HT$_{1B}$* mutants is due to 5-HT$_{1B}$ malfunction in hemocytes.

To investigate the role of 5-HT$_{2B}$ in *Drosophila* hemocyte phagocytosis, we also knocked down *5-HT$_{2B}$* expression in blood cells by RNAi using two independent *UAS-5-HT$_{2B}$* RNAi lines. Unlike the results found in *P. rapae*, the reduced levels of *5-HT$_{2B}$* in *Drosophila* hemocytes led to a defect in the phagocytosis of *E. coli* and *S. aureus* (*Figure 8A–D*). Moreover, the flies expressing *5-HT$_{2B}$* RNAi in their hemocytes took up significantly fewer latex beads than did control flies after 5-HT injection (*Figure 8E*). 5-HT$_{2B}$ knockdown flies had higher bacterial loads than controls (*Figure 8I*) and increased susceptibility to *S. aureus* (*Figure 8E–G*). Taken together, our experiments demonstrate that both 5-HT$_{1B}$ and 5-HT$_{2B}$ play important roles in host defense against bacterial infection.

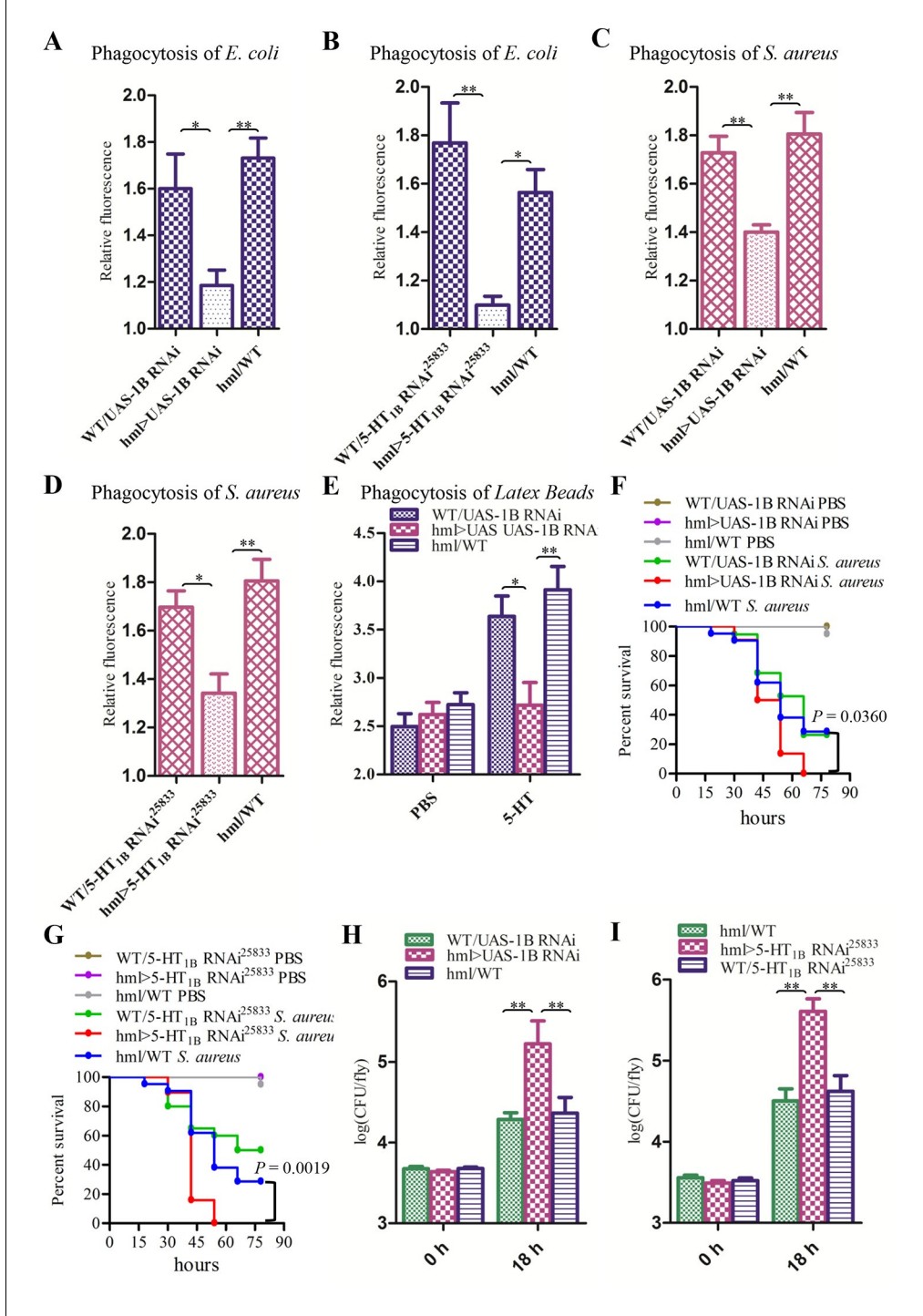

**Figure 7.** Knockdown of 5-HT$_{1B}$ in hemocytes affects *Drosophila* phagocytosis and survival. (**A**) Quantification of in vivo phagocytosis of *E. coli* in WT/*UAS-1B RNAi*, *hml> UAS-1B RNAi* and *hml/*WT flies. Approximately six flies per genotype were used in each experiment. Experiments were performed twice, (**B**) Quantification of in vivo phagocytosis of *E. coli* in WT/*UAS-5-HT$_{1B}$ RNAi$^{25833}$*, *hml>UAS-5-HT$_{1B}$ RNAi$^{25833}$*, and *hml/*WT flies. Approximately six flies per genotype were used in each experiment. Experiments were performed twice. (**C**) Quantification of in vivo phagocytosis of *S. aureus* in WT/*UAS-1B RNAi*, *hml> UAS-1B RNAi,* and *hml/*WT flies. Approximately six flies per genotype were used in each experiment. Experiments were performed twice, (**D**) Quantification of in vivo phagocytosis of *S. aureus* in WT/*UAS-5-HT$_{1B}$ RNAi$^{25833}$*, *hml>UAS-5-HT$_{1B}$ RNAi$^{25833}$*, and *hml/*WT flies. Approximately six flies per genotype were used in each experiment. Experiments were performed twice. (**E**)
*Figure 7 continued on next page*

*Figure 7 continued*

Quantification of in vivo phagocytosis of red fluorescently labeled latex beads in WT/*UAS-1B RNAi, hml>UAS-1B RNAi*, and *hml*/WT flies after a 30 min preinjection of either PBS or 1µg/µl 5-HT. Approximately six flies per genotype were used in each experiment. Experiments were done twice, (**F**) Representative survival curves of WT/*UAS-1B RNAi, hml>UAS-1B RNAi*, and *hml*/WT male flies after injection of *S. aureus*. n=19–22 flies. Data are *representatives* of three independent experiments. Each experiment was performed in triplicate. (**G**) Representative survival curves of WT/*UAS-5-HT$_{1B}$ RNAi$^{25833}$, hml>UAS-5-HT$_{1B}$ RNAi$^{25833}$*, and *hml*/WT male flies after injection of *S. aureus*. n=19–21 flies. Data are *representatives* of two independent experiments. Each experiment was performed in triplicate. (**H**) Comparison of the *S. aureus* (OD 0.4) recovered in WT/*UAS-1B RNAi, hml>UAS-1B RNAi* and *hml*/WT flies 0, and 18 hr post infection. Bacterial load was measured in eight individual male flies per genotype at each time point in each experiment. Experiments were performed in triplicate. (**I**) Comparison of the *S. aureus* (OD 0.4) recovered in WT/*UAS-5-HT$_{1B}$ RNAi$^{25833}$, hml>UAS-5-HT$_{1B}$ RNAi$^{25833}$*, and *hml*/WT flies 0 and 18 hr post infection. Bacterial load was measured in eight individual male flies per genotype at each time point in each experiment. Experiments were performed in triplicate. One-way ANOVA followed by Tukey's multiple comparison test for **A**, **B**, **C**, **D**, **E**, **H**, and **I**. Error bars indicate ± s.e.m., ***p<0.001, **p<0.01, *p<0.05.

## Discussion

Our study demonstrates the ubiquity of serotonergic receptor signaling pathway in immune function. This paper is the first to demonstrate that insect hemocytes express TPH and can synthesize 5-HT, like human macrophages (*Nakamura et al., 2008*). The exposure of hemocytes to LPS led to an induction of TPH expression and a release of 5-HT. Moreover, the secreted 5-HT appears to be an important autocrine stimulus. It promotes phagocytosis: inhibition of TPH either by a competitive inhibitor or siRNA silencing resulted in significantly decreased hemocyte phagocytosis.

Chemicals that act as neurotransmitters in the nervous system can also modulate immune function (*Meredith et al., 2005*). 5-HT is one of these classical neurotransmitters that is also an important immune regulatory molecule in both insects and mammals. Recent research has made progress in determining 5-HT modulated mammalian immune responses, especially regarding the machinery to produce, store, release, and respond to 5-HT in immune cells (*Ahern, 2011*). 5-HT was also reported to mediate immune responses, such as hemocyte phagocytosis, nodule formation and hemocyte population in insects (*Baines et al., 1992; Kim et al., 2009; Kim and Kim, 2010*), but the signaling pathway is unclear. *Drosophila* TPH homolog gene *Henna* was found to be one hit in a genome-wide RNAi screen for genes that affect phagocytosis of *Candia albicans* by hemocytes (*Stroschein-Stevenson et al., 2006*). Examining the basic interactions between serotonin receptor-medicated second messenger systems and immune-related intracellular signaling pathways in insects may shed light on their interactions and functions in mammals.

Second, our findings show that naive hemocytes express 5-HT$_{1B}$, 5-HT$_{2B}$, and 5-HT$_7$ receptors, but only 5-HT$_{1B}$ and 5-HT$_{2B}$ appear to have functional roles. Using selective antagonists and RNAi, we found that inhibition of 5-HT$_{1B}$ decreases hemocyte phagocytosis. However, inhibition of 5-HT$_{2B}$ enhances hemocyte phagocytosis. It seems that activation of these two receptors affects hemocytes in opposite ways. Interestingly, we found that 5-HT$_{1B}$ is dramatically up-regulated following hemocyte activation, but 5-HT$_{2B}$ is significantly down-regulated. Therefore, even though the two receptors have opposite functions upon activation, the overall effects of 5-HT signaling induced by immune challenge are favorable for hemocyte phagocytosis (*Figure 7*). Mammalian DCs also express 5-HT$_{1B}$ and 5-HT$_{2B}$, via which 5-HT induces chemotaxis and in vivo migration of DCs to draining lymph nodes (*Müller et al., 2009*). Even though insect plasmatocytes are macrophage-like cells, 5-HT activates macrophage cells via 5-HT$_{1A}$ (*Nakamura et al., 2008*) and 5-HT$_{2C}$ (*Mikulski et al., 2010*), which are different from the receptor sub-types found in insect hemocytes. The 5-HT$_7$ receptor plays a critical role in the immune response in the gut of mice (*Kim et al., 2013*). Although our results indicated that 5-HT$_7$ receptors were not involved in insect hemocyte phagocytosis, they may be critical for other immune activities.

Many neurotransmitter receptors are found on mammalian immune cells and regulate innate immune response, however, it is still unclear about their general role at the organismal level because most studies are conducted in vitro (*Sternberg, 2006*). However, it is relatively easy to test immunity in insects in vivo (*Lemaitre and Hoffmann, 2007*). We found that the *5-HT$_{1B}$* deficiency flies were

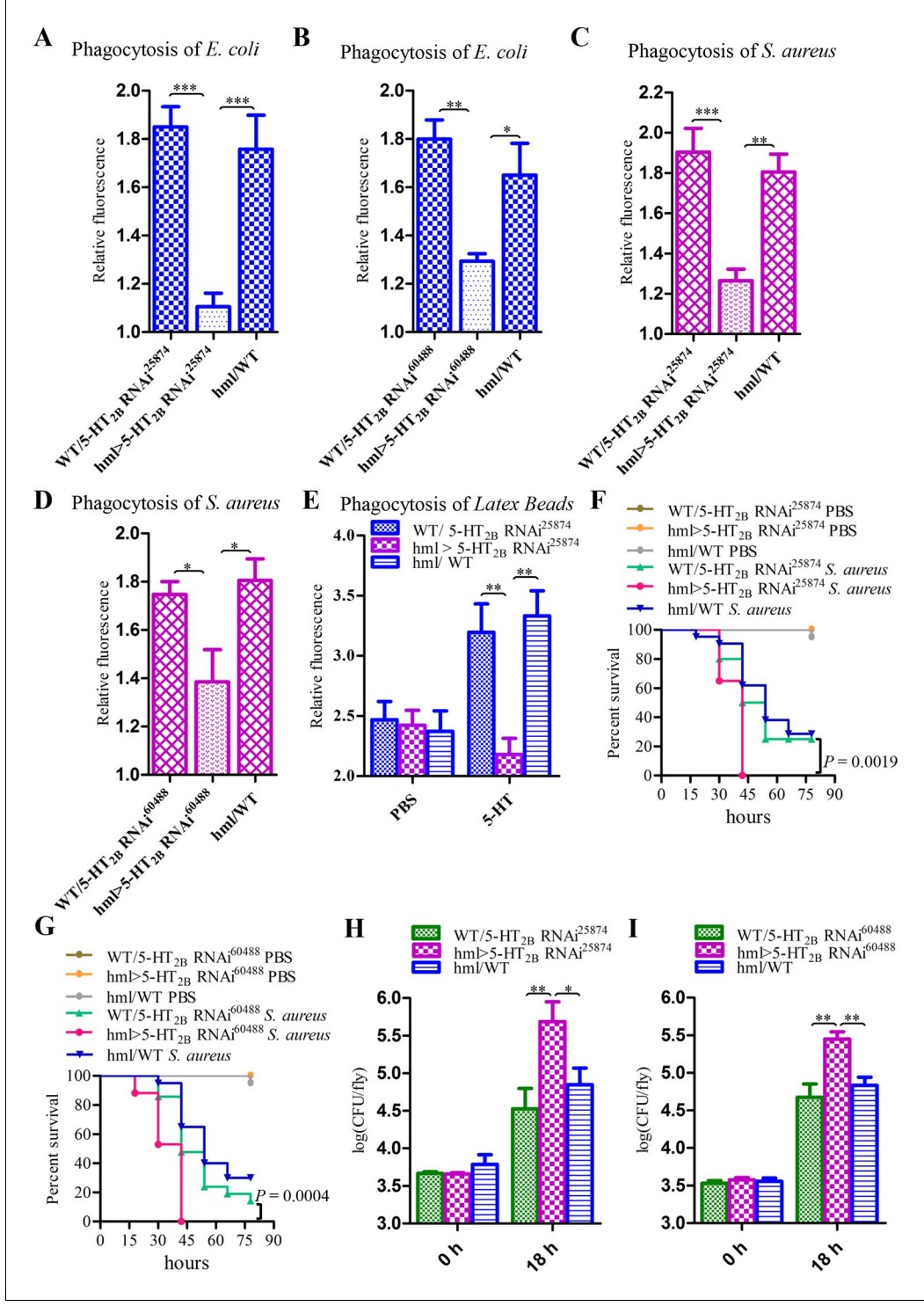

**Figure 8.** Knockdown of 5-HT$_{2B}$ in hemocytes affects *Drosophila* phagocytosis and survival. (**A**) Quantification of in vivo phagocytosis of *E. coli* in WT/*UAS-5-HT$_{2B}$ RNAi$^{25874}$*, *hml>UAS-5-HT$_{2B}$ RNAi$^{25874}$*, and *hml/*WT flies. Approximately six flies per genotype were used in each experiment. Experiments were performed twice, (**B**) Quantification of in vivo phagocytosis of *E. coli* in WT/*UAS-5-HT$_{2B}$ RNAi$^{60488}$*, *hml>UAS-5-HT$_{2B}$ RNAi$^{60488}$*, and *hml/*WT flies. Approximately six flies per genotype were used in each experiment. Experiments were performed twice. (**C**) Quantification of in vivo phagocytosis of *S. aureus* in WT/*UAS-5-HT$_{2B}$ RNAi$^{25874}$*, *hml>UAS-5-HT$_{2B}$ RNAi$^{25874}$*, and *hml/*WT flies. Approximately six flies per genotype were used in each experiment. Experiments were performed twice, (**D**) Quantification of in vivo phagocytosis of *S. aureus* in WT/*UAS-5-HT$_{2B}$ RNAi$^{60488}$*, *hml>*

*Figure 8 continued on next page*

*Figure 8 continued*

UAS-5-HT$_{2B}$ RNAi$^{60488}$, and *hml*/WT flies. Approximately six flies per genotype were used in each experiment. Experiments were performed twice. (**E**) Quantification of in vivo phagocytosis of red fluorescently labeled latex beads in WT/UAS-5-HT$_{2B}$ RNAi$^{25874}$, *hml*> UAS-5-HT$_{2B}$ RNAi$^{25874}$, and *hml*/WT flies after a 30 min preinjection of either PBS or 1µg/µl 5-HT. Approximately six flies per genotype were used in each experiment. Experiments were done twice. (**F**) Representative survival curves of WT/UAS-5-HT$_{2B}$ RNAi$^{25874}$, *hml*> UAS-5-HT$_{2B}$ RNAi$^{25874}$, and *hml*/WT male flies after injection of *S. aureus*. n=20–21 flies. Data are *representatives* of two independent experiments. Each experiment was performed in triplicate. (**G**) Representative survival curves of WT/UAS-5-HT$_{2B}$ RNAi$^{60488}$, *hml*>UAS-5-HT$_{2B}$ RNAi$^{60488}$, and *hml*/WT male flies after injection of *S. aureus*. n=19–21 flies. Data are *representatives* of two independent experiments. Each experiment was performed in triplicate. (**H**) Comparison of the *S. aureus* (OD 0.4) recovered in WT/UAS-5-HT$_{2B}$ RNAi$^{25874}$, *hml*>UAS-5-HT$_{2B}$ RNAi$^{25874}$, and *hml*/WT flies 0, and 18 hr post infection. Bacterial load was measured in eight individual male flies per genotype at each time point in each experiment. Experiments were performed in triplicate. (**I**) Comparison of the *S. aureus* (OD 0.4) recovered in WT/UAS-5-HT$_{2B}$ RNAi$^{60488}$, *hml*>UAS-5-HT$_{2B}$ RNAi$^{60488}$, and *hml*/WT flies 0, and 18 hr post infection. Bacterial load was measured in eight individual male flies per genotype at each time point in each experiment. Experiments were performed in triplicate. One-way ANOVA followed by Tukey's multiple comparison test for **A**, **B**, **C**, **D**, **E**, **H**, and **I**. Error bars indicate ± s.e.m., ***$p<0.001$, **$p<0.01$, *$p<0.05$.

more vulnerable to bacterial infections due to their poor phagocytic ability. Hemocyte-specific RNAi experiments showed similar results, indicating that the 5-HT$_{1B}$-mediated hemocyte phagocytosis is important for insects to defend themselves against pathogens. Surprisingly, the flies expressing 5-HT$_{2B}$ RNAi in their hemocytes were more susceptible to bacterial infections, suggesting that activation of 5-HT$_{2B}$ may promote phagocytosis in *Drosophila*, different from the results in *P. rapae*.

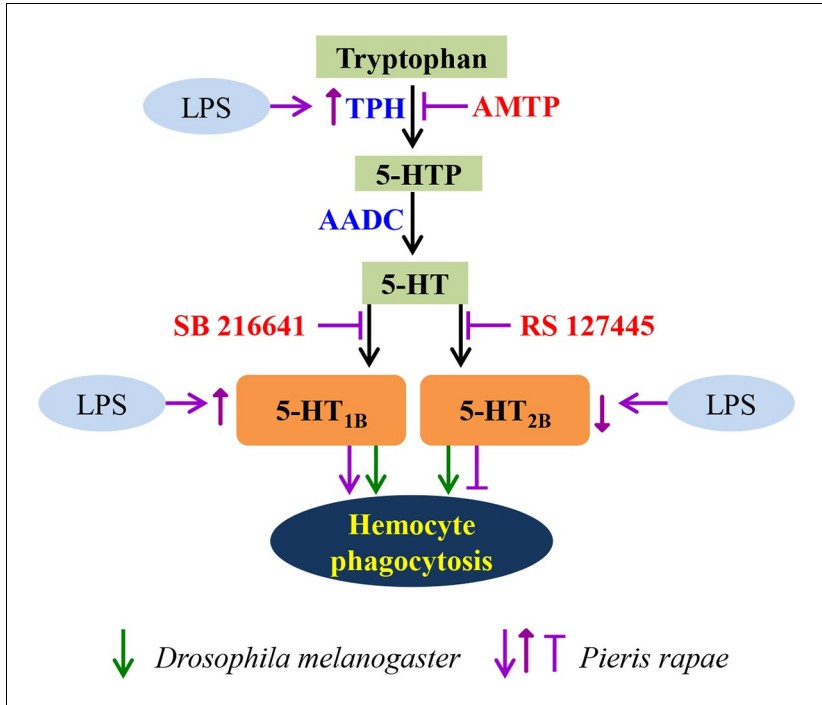

**Figure 9.** A schematic diagram of serotonin signaling on hemocyte phagocytosis. LPS enhances the expression of TPH, which catalyzes tryptophan into 5-HT via 5-HTP. 5-HT, which secreted from hemocytes, activates the hemocyte-membrane receptor 5-HT$_{1B}$ and 5-HT$_{2B}$. The immune responses of *P. rapae* are labeled in purple: activation of 5-HT$_{1B}$ promotes hemocyte phagocytosis and activation of 5-HT$_{2B}$ lead to opposite effects. LPS increases 5-HT$_{1B}$ expression but decreases that of 5-HT$_{2B}$. The immune responses of *Drosophila* are labeled in green arrows: activation of 5-HT$_{1B}$ promotes hemocyte phagocytosis and activation of 5-HT$_{2B}$ lead to the same effects.

In conclusion, we found that insect hemocytes can synthesis and release 5-HT, which regulates phagocytosis via $5\text{-HT}_{1B}$ and $5\text{-HT}_{2B}$ receptors on the membrane of the hemocyte. We also used the genetic model *Drosophila* to further confirm the roles of $5\text{-HT}_{1B}$ and $5\text{-HT}_{2B}$ in vivo. These findings suggest that serotonin, an ancient signaling molecule, modifies immune function in animals across phyla. Exploring the interactions between serotonin receptor-mediated pathways and immune-related pathways may be easier to initiate in insects, which have a simpler immune system.

## Materials and methods

### Insects

*P. rapae* larvae were collected primarily from cabbage fields in the experimental farmland of Zhejiang University, Hangzhou, China and the laboratory colony of *P. rapae* was reared at 25 ± 1°C and 70% relative humidity under a photoperiod of 14 hr light: 10 hr dark in a greenhouse as previously described by *Zhang et al. (2005)*.

The following fly stocks were from the Bloomington Drosophila Stock Center: *HmlΔ-Gal4* (30139); *UAS-5-HT$_{1B}$ RNAi* (27634, 25833); *UAS-5-HT$_{2B}$ RNAi* (25874, 60488). The *5-HT$_{1B}$* control and *5-HT$_{1B}$$^{ΔIII-V}$* flies were generous gifts from Leslie Vosshall (*Gasque et al., 2013*).

### Reagents

Serotonin hydrochloride, forskolin, G418 disulfate salt and SB-269970 hydrochloride were obtained from Sigma-Aldrich (St Louis, MO). SB 216641 and RS 127445 were purchased from Tocris Bioscience (Bristol, UK).

### Hemocytes isolation, culture and immune induction

Fifth-instar larvae of *P. rapae* were surface sterilized with 70% ethanol. The proleg was cut with a pair of scissors and the hemolymph was collected in Grace's Insect Medium (1:10, v/v; Invitrogen, Carlsbad, CA). The diluted hemolymph was added to 12-well tissue culture plates (Nunc, Roskilde, Denmark) at a density of $2 \times 10^6$ cells per well. Hemocytes were treated with 100 ng/ml LPS (*Escherichia coli* 0111:B4; Sigma Aldrich) for 15 min, 1 hr, 2 hr, and 4 hr (*Ngkelo et al., 2012*; *Wu et al., 2015*). The hemocyte culture was collected and centrifugated. The supernatants were collected for 5-HT detection. The hemocytes that adhered to the plate were harvested for RNA isolation.

### 5-HT detection

The hemolymph from three first day of fifth instar larvae was collected and mixed. The combined hemolymph (30 μl) was mixed with 170 μl Grace's Insect Medium (Invitrogen) containing 50 μg/ml tetracycline and 2 μl saturated 2-phenylthiourea (PTU). The diluted hemolymph was added to each well of a 8-well chambered coverglasses (Lab-Tek, Nunc, Thermo Fisher Scientific, Rochester, USA) and hemocytes were allowed to adhere to the slide for 20 min at 27°C to form monolayers. Then, hemocytes were fixed with 4% paraformaldehyde. 5-HT was labeled with 5-HT antisera (72 hr, room temperature; S-5545; Sigma) followed by biotinylated anti- rabbit Ig (24 hr, 4°C) and SA-Alexa Fluor 546 (1 hr, room temperature; Invitrogen). For use as a negative control, 5-HT antisera was preabsorbed with 5-HT (10 mM, 24 hr, 4°C). The nuclei of hemocytes were stained with 1 μg/ml of 4'-6-diamidino-2-phenylindole (DAPI, Beyotime Biotech, Jiangsu, China) for 5 mins and hemocytes were observed by fluorescent microscope (Zeiss, Göttingen, Germany).

We performed ELISA to quantify the amount of 5-HT produced in the hemocyte culture supernatants using the 5-Hydroxytryptamine (serotonin ) assay kit (*Li et al., 2014*) (Jiancheng, Nanjing, China).

### Cloning of 5-HT receptors, TRH and TPH

Total RNA was isolated from *P. rapae* nerve cord with Trizol reagent (Invitrogen, Carlsbad, CA, USA). Single-strand cDNA, synthesized from the RNA using a ReverTra Ace-α- kit (Toyobo, Osaka, Japan), was used as a template for PCRs. We performed transcriptome sequencing of nerve cord of *P. rapae*. Through transcriptome sequencing, several putative serotonin receptors were annotated using BlastX (National Center for Biotechnology Information [NCBI], Bethesda, MD). The full length was obtained using the 5'-Full rapid-amplification of cDNA ends (RACE) Kit (Takara, Dalian, China)

and 3'-Full RACE Kit (Takara, Dalian, China) (*Supplementary file 1A*). To amplify the complete sequence, we use the forward primer located upstream of the putative start codon initiator, and the reverse primer located downstream of the putative stop codon (*Supplementary file 1A*).

## qPCR and RT-PCR

Total RNA was extracted from *P. rapae* hemocytes with high pure RNA isolation kit (Roche) in accordance with the manufacturer's instructions. To collect RNA from larval fly blood cells, approximately 30 larvae were carefully lacerated with tweezers on their anterior end in 100 µl of nuclease-free water. The RNA quantity and quality was measured by using a *Nanodrop 2000 spectrophotometer* (Thermo Scientific Inc., Bremen, Germany). Reverse transcription was performed with 1 µg of RNA by using a ReverTra Ace qPCR RT kit (Toyobo, Osaka, Japan). For conventional reverse transcription-polymerase chain reaction (RT-PCR) cDNA was amplified using KOD- Plus- (Toyobo, Osaka, Japan). Real-time quantitative PCR was performed on cDNA preparations using the SsoFast Eva Green Supermix with Low Rox (Bio-Rad, Hercules, CA) and Applied Biosystems 7500 Real-Time PCR System (Applied Biosystems by Life Technologies, Carlsbad, CA) following the manufacturer's instructions. The quantification of transcript level of different gene was conducted according to the $2^{-\Delta\Delta CT}$ method (*Livak and Schmittgen, 2001*). Comparable quantities of cDNA were ensured by amplification of 18s rRNA, as a stably expressed reference gene (*Wu et al., 2013*) in *P. rapae*. The primers are listed in *Supplementary file 1A*.

## Phagocytosis assay

### Hemocyte phagocytosis assays in vitro

The assay for phagocytosis was performed according to the method described by *Cuttell et al., (2008)*, with minor modifications. After isolation of hemocytes, they were prepared in a 96-well tissue culture plate (CoStar, Washington). Cells were incubated for 1 hr counterstained with 20 µM Cell Tracker Blue CMAC (Molecular Probes) (Life Technologies, Carlsbad, CA) and then washed in PBS. 5 µl of the drug solution were mixed with 45 µl of Grace's medium were added into each well for 30 min. Bacterial phagocytosis assays using pHrodo *E. coli* (Invitrogen, Carlsbad, CA) were performed according to the manufacturer's instructions. The proportion of cells that had phagocytosed labeled *E. coli* was determined under a florescence microscope (Nikon Eclipse TS100, Nikon, Japan) at 200 × in five different fields.

## Fly phagocytosis assays in vivo

To assay *E. coli* and *S. aureus* phagocytosis in adults, approximately eight to ten flies with an equal distribution of females and males per genotype were injected with ~0.2 µl of 1 mg/mL pHrodo green *E. coli* or pHrodo red *S. aureus* (Invitrogen, Carlsbad, CA) using a FemtoJet micro-injection system (Eppendorf). They were then incubated for 1 hr at room temperature. Fluorescently labeled particles were visualized through the cuticle using a florescence microscope (Nikon AZ100M, Nikon, Japan). We use Image J software to quantify the results. Relative fluorescence calculated as: $[\text{fluorescence}]_{\text{dorsal vein area}}/[\text{fluorescence}]_{\text{adjacent area}}$.

To assay phagocytosis of beads, flies were first injected with approximately 36.8 nl PBS or 1 µg/µl 5-HT dissolved in PBS using a Drummond Scientific Nanoject II. Flies were incubated at room temperature for 30 min and then injected with approximately 27 nl of 1.0 µm Red Fluorescent Carboxylate Modified FluoSpheres diluted 1: 2 (Invitrogen), incubated at room temperature for 10 min, injected with 36.8 nl 0.4% Trypan blue (Invitrogen), and then mounted and visualized as described above.

## Hemocytes RNAi

Small interfering RNA (siRNA) molecules were designed from the nucleotide sequence obtained from *P. rapae* using Invitrogen siRNA design software (http://rnaidesigner.thermofisher.com/rnaiexpress/) (*Supplementary file 1B*) and were synthesized by Invitrogen. siRNA for the negative control was used in transfection experiments as a control. Hemocytes were isolated as described above and were attached to the surface of 96-well tissue culture plates and incubated in Grace's medium. The monolayers of hemocytes were treated with 2.4 ng siRNA and 0.7 µl siRNA transfection reagent

INTERFERin (Polyplus-transfection SA, France) according to the instructions from the manufacturer, and incubated at 27°C.

## Polyclonal antibody preparation

We use the forward primer Pr5-HT$_{1B}$- BamHI 5′- CG*GGATCC*CAAACAGCTAGGAAAAGAAT -3′ and the reverse primer Pr5-HT$_{1B}$- XhoII 5′- CC*CTCGAG*TTATGTCTTTGCCGCTTTCC -3′ to amplify the third intracellular loop cDNA fragment of Pr5-HT$_{1B}$. Purified PCR products was cloned into vector pET-28a (+) and confirmed by DNA sequencing. The expression product was 17 kDa which carried a His-tag. The construct was used to transform *Escherichia coli* BL21 (DE3) and the cells inoculated into 2 l of Luria–Bertani (LB) medium containing kanamycin (50 mg/ml) at 37°C. Until A600 reached 0.8, the cultures were added with 0.5 mM isopropyl-β-d-thiogalactopyranoside (IPTG) and incubated overnight at 28°C. The cells were collected by centrifugation and disrupted by sonication. The insoluble recombinant protein was purified through Ni-chelating affinity column (TransGen Biotech, Beijing, China) under denaturing conditions. To confirm the identity of the recombinant protein, proteins were separated by SDS-PAGE, transferred to PVDF (Bio-Rad) and detected with an anti-His polyclonal antibody HRP (HuaAn Biotechnology, Hangzhou, China) conjugate by ECL (Thermo Scientific). Then the purified protein was submitted to Hangzhou HuaAn Biotechnology Company as an antigen for immunization in male New Zealand rabbits. The antibody to Pr5-HT$_{1B}$ was purified from antiserum which had been done by the company.

Polyclonal antibody against Pr5-HT$_{2B}$ was generated by Hangzhou HuaAn Biotechnology Company using peptide antigen for antibody development. Epitopes are predicted by GenScript Optimum Antigen design tool. Peptide sequence synthesized for Pr5-HT$_{2B}$ antibody development is SAAAKTSKGTNISEC. The amino acid cysteine (C), which locates at the end of the peptide, is used for carrier protein conjugation.

## Western blot and immunofluorescence

Hemocytes were lysed in ice-cold buffer (1% Triton X-100, 150 mM NaCl, 10 mM Tris-HCl pH 7.5, 5 mM EDTA, 1 mM sodiumo-vanadate) containing protease inhibitors phenylmethyl sulfonylfluoride (PMSF, Sangon Biotech, P0754, Shanghai, China). For membrane proteins, debris was sedimented by centrifugation (15,500 g, 10 min, 4°C), supernatants were collected and their protein concentration was determined by Bradford reagent (Sangon Biotech, Shanghai, China). Samples were diluted in 4 × Protein SDS PAGE Loading Buffer (Takara Biotechnology, Japan), then boiled for 10 min. Samples were separated in a denaturing polyacrylamide gel and transferred to a PVDF membrane. After blocking (5% Tris-buffered saline; pH 7.0; containing 0.1% Tween 20) and washing, membranes were then incubated overnight with primary antibodies against Pr5-HT$_{1B}$ and Pr5-HT$_{2B}$ (1: 500). Membranes were then incubated with secondary antibody horseradish peroxidase–conjugated goat anti-rabbit IgG diluted 1:5,000 in Tris-buffered saline with Tween-20. Membranes were rinsed five times with wash buffer and then incubated with the ECL western blotting substrate (Promega, Wisconsin). Membranes were stripped and reprobed with anti-β- actin (Cell Signaling Technology, Beverly, MA) for 2 hr at room temperature, followed by incubation with secondary antibody.

Immunofluorescence was performed using the same methods as used for 5-HT detection except that the primary antibody was different.

## Construction of expression plasmids

An expression plasmid containing the Kozak consensus sequence (*Kozak, 1987*) was constructed by PCR with specific primers (*Supplementary file 1A*). The PCR product was double digested. The digested DNA fragments were purified by PCR Clean-Up Kit (Axygen, Union City, CA) and then subcloned into pcDNA3.0 vector (Invitrogen). The correct insertion was confirmed by DNA sequencing.

## HEK 293 cell culture, transfection, and creation of stable cell lines

Human Embryonic Kidney 293 (HEK 293) cells were obtained from the Cell Bank of Type Culture Collection of Chinese Academy of Sciences (Shanghai, China). The cell line identity has been authenticated by short tandem repeat (STR) profiling, and the cells were free from the mycoplasma contamination. HEK-293 cells were grown in Dulbecco's modified Eagle's medium (D-MEM) (Gibco

BRL, Gaithersburg, MD) supplemented with 10% fetal bovine serum (FBS) (Gibco BRL) and antibiotics at 37°C and 5% $CO_2$. After transfection of plasmid into the cells using Lipofectamine 2000 (Invitrogen), the antibiotic G418 (0.8 mg/mL) was added to the medium to select for cells that stably expressed the receptors. After 2 weeks of G418 selection, G418- resistant colonies were trypsinized in cloning cylinders and transferred to 12-well plastic plates for expansion. These individual cell lines were analyzed for integration of the receptor DNA by RT-PCR and localization of the protein by immunofluorescence (data not shown). The clonal cell line that most efficiently expressing 5-HT receptor was chosen for this study.

## cAMP assay

Intracellular cAMP concentration ([cAMP]$_i$) was determined as previously described (*Huang et al., 2007*). Cells were plated into 12-well tissue culture plates (Nunc, Roskilde, Denmark) at a density of $1 \times 10^6$ cells per well and incubated at 37°C with 10% $CO_2$ in a humidified incubator. The cells were pre-incubated in Dulbecco's phosphate-buffered saline (DPBS; Gibco-Invitrogen) containing 100 uM phosphodiesterase inhibitor IBMX for 20 min at room temperature. After the preincubation, a 50-µl aliquot of D-PBS containing various concentrations of reagents was added. The culture was then incubated for 20 min at room temperature. The reaction was stopped by aspirating the solution and then adding 250 µl ice-cold cell lysis buffer immediately to lysis the cells. The cell lysate was scraped into 1.5 ml Eppendorf tubes for collections and stored at –70°C until use. The solution was centrifugated and [cAMP]$_i$ in the supernatant was determined using a cAMP ELISA kit (R&D Systems, Minneapolis, MN).

## Ca$^{2+}$ imaging

Intracellular calcium concentration ([Ca$^{2+}$]$_i$) was estimated and analyzed with a calcium imaging system. Fresh hemocytes from fifth-instar larvae were seeded on the coverslip with Grace's medium and incubated for 30 min at 27°C. Then, the cells were loaded with the fluorescent probe Fura 2-AM (Dojindo Laboratories, Kumamoto, Japan) using 0.2% Cremophor EL (Sigma–Aldrich) for 30 min at 27°C. The cells were subsequently washed twice with a bathing solution (152 mM NaCl, 5.4 mM KCl, 5.5 mM glucose, 1.8 mM CaCl$_2$, 0.8 mM MgCl$_2$, and 10 mM HEPES, pH 7.4). The coverslips were transferred to a microscopic chamber that was constantly perfused with the bathing solution at ≈ 2 ml/min (*Huang et al., 2012*). The fluorescence at 510 nm by excitation at 340 or 380 nm with a xenon lamp was measured with individual cells using an Easy Ratio Pro calcium imaging system (Photon Technology International, Birmingham, NJ). Each experiment was repeated three times or more. EC$_{50}$ values were estimated by fitting to a dose-dependent curve in Origin Pro (Origin Lab, Northampton, MA).

## Survival following infection

The tetracycline-resistant *S. aureus* was grown overnight in a shaking incubator at 37°C shaking at 225 rpm. Cultures were spun down. The resuspended cells were diluted in sterile PBS to achieve an optical density (OD) of 0.4. Five to seven days post-eclosion, flies were injected with 32.2 nl of the bacterial resuspension using a Drummond Scientific Nanoject II. Flies were kept at 25°C. Flies that died within 6 hr were considered dead and were removed from the count. Flies were transferred every day to new food and death was recorded every 12 hr.

## Bacterial load

The tetracycline-resistant *S. aureus* was cultured in LB broth overnight at 37°C shaking at 225 rpm, and subcultured to an OD of 0.4. Approximately 24 female flies per genotype were injected with 23 nl of the bacterial suspension. Eight flies from each group were then immediately homogenized in individual tubes with 100 µl sterile PBS. The material was serially diluted to 1:10, twice, in sterile PBS, and then plated on LB plates containing tetracycline. After 6 and 18 hr of infection at 25°C, eight additional flies from each genotype per time point were assayed as above with the exception that each sample was serially diluted 1:10 five times. The plates were incubated at 37°C for 8 to 10 hr. Bacterial colonies were counted when the colonies remain small and discrete.

## Statistical analysis

Data had a normal distribution and are expressed as means ± standard error (s.e.m.). Data were analyzed using analysis of variance (ANOVA) with Tukey-Kramer post-hoc test and Student t tests. Log rank tests were used to determine whether survival curves of male flies were significantly different from one another. Gehan-Breslow-Wilcoxon test were used to determine whether survival curves of female flies were significantly different from one another. All curve fitting and statistical calculations were performed with Origin 8.0 (Origin Lab, Northampton, MA) and GraphPad Prism 5.0 (San Diego, CA).

## Acknowledgements

This work was supported by National Program on Key Basic Research Projects (2013CB127600), National Natural Science Foundation of China (31572039) and China National Science Fund for Innovative Research Groups of Biological Control (31321063). We thank Leslie Vosshall (The Rockefeller University) for sharing $5\text{-}HT_{1B}$ control and $5\text{-}HT_{1B}^{\Delta III\text{-}V}$ flies and Yu Yun-song (College of Medicine, Zhejiang University) for providing the bacterial strain *Staphylococcus aureus*. We thank Shelley Adamo (Dalhousie University) for her helpful comments on the manuscript.

## Additional information

### Funding

| Funder | Grant reference number | Author |
| --- | --- | --- |
| Ministry of Science and Technology of the People's Republic of China | 2013CB127600 | Jia Huang |
| National Natural Science Foundation of China | 31572039 | Jia Huang |
| National Natural Science Foundation of China | 31321063 | Gong-yin Ye |

The funders had no role in study design, data collection and interpretation, or the decision to submit the work for publication.

### Author contributions

Y-xQ, Performed experiments, Helped write the paper; JH, Designed the study, Analyzed the data, Wrote the paper; M-qL, Y-sW, R-yX, Helped perform experiments; G-yY, Organized the project

## Additional files

### Supplementary files

• Supplementary file 1. Primers used in this research article and siRNA sequence are presented.

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
