## [Decision Letter]

Thank you for submitting your work entitled "Serotonin modulates insect hemocyte phagocytosis via two different serotonin receptors" for consideration by *eLife*. Your article has been favorably evaluated by three reviewers, including Dan Hultmark and a member of our Board of Reviewing Editors, and the evaluation was overseen by K VijayRaghavan as the Senior editor.

The reviewers have discussed the reviews with one another and the Reviewing Editor has drafted this decision to help you prepare a revised submission.

Summary:

In Qi et al., the authors present evidence that the neurotransmitter serotonin is secreted by activated hemocytes in the caterpillar *Pieris rapae*. Inhibition of serotonin production decreased hemocyte phagocytosis. The authors identified two serotonin receptors expressed in hemocytes, *Pr5-HT_1B_* and *Pr5-HT_2B_*. Blocking the activity of each of these receptors demonstrates that *Pr5-HT_1B_* promotes phagocytosis, while *Pr5-HT_2B_* inhibits it. However, upon activation by LPS, the authors observed that *Pr5-HT_1B_* expression is up-regulated while *Pr5-HT_2B_* is down-regulated, resulting in a net increase in phagocytosis during activation. The authors wanted to test the if inhibition of serotonin signaling decreases hemocyte phagocytosis in vivo, so they performed bacterial infection assays in *Drosophila melanogaster*, comparing wild-type flies to 5-HT_1B_ receptor mutants. They found decreased phagocytosis of both gram-negative and gram-positive bacteria in 5-HT_1B_ receptor mutants as well as decreased survival upon infection. Overall, the authors argue that serotonin signaling plays an important role in insect innate immunity.

Essential revisions:

The work done in the caterpillar is well done and nicely controlled. There are similar results already published for other insect species (Kim GS, Nalini M, Kim Y, Lee DW, Arch Insect Biochem Physiol. 2009; Kim GS and Kim Y, J Insect Physiol. 2010) and these need to be acknowledged up front in the introductory section with a sentence justifying why this study is still relevant. The reviewers found that the butterfly experiments are more detailed than in the older literature and that the extension to *Drosophila* opens up new possibilities for extending this research. Unfortunately, it is the fruit fly results that are not substantial enough to warrant an *eLife* publication. If some more characterization is presented in terms of phenotype and mechanisms, then this will achieve that goal. The authors are the best judges of the experiments that will further consolidate the fruit fly story. The manuscript will benefit from such addition. Of course, if necessary changes will take longer than 2 months, the authors could resubmit an expanded manuscript. The following major questions are provided as examples of questions that need to be addressed.

1) Is the receptor 5-HT_1B_ required autonomously in hemocytes?

Does hemocyte-autonomous RNAi of 5-HT_1B_ affect the phagocytosis response, CFU after infection, and overall survival after infection? The authors could use *Hml∆-GAL4* and ideally two independent *UAS*-RNAi lines. For suitable readouts of CFU and phagocytosis see below.

2) Does 5-HT_1B_ indeed mediate phagocytic behavior in hemocytes and bacterial clearance as measured by CFU (or other hemocyte response/s)?

Regarding the quantification of CFU after bacterial infection, it is unclear if bacterial strains with a resistance gene were used? This is not mentioned in the methods and needs to be clarified. Use of bacteria with a resistance marker is needed so injected bacteria can be distinguished from the naturally occurring bacterial load.

To carefully evaluate phagocytosis the authors should, at a minimum, use the trypan blue phagocytosis assay as described by Elrod-Erickson et al. Curr. Biol. 2000.

3) Does infection or LPS challenge indeed trigger an increased phagocytic hemocyte response through 5-HT_1B_?

Does hemocyte-specific kd of 5-HT_1B_ dampen phagocytosis in general, or is there indeed a lack of inducibility, e.g. after LPS challenge? Comparison of injecting fluorescent beads -/+LPS would allow to address this. A wt control should be used in parallel to confirm inducibility by the immune challenge here. The experiment could also be performed by injecting fluorescent beads with -/+ 5-HT.

4) Do 5-HT levels increase upon bacterial infection in *Drosophila*? It would be easy to perform a hemolymph ELISA (as in Figure 1) to see if 5-HT levels increase upon infection in flies.

5) Both 5-HT_1B_ and 5-HT_2B_ seem to be expressed in hemocytes of uninfected adult *Drosophila* (Figure 6). But the entire focus of the infection study phenotype is limited to the 1B mutant. What effects, if any, does inhibition/reduction of 2B have on phagocytosis and the lifespan of immune challenged animals? Would it be the same as in the caterpillar? If there is no mutant for 2B, the authors could minimally check levels of 2B expression in the hemocytes of infected animals, and perhaps one would see a down regulation similar to that in the caterpillar. There are MiMIC and TRiP lines available at BDSC, and minimally, 2B RNAi lines could be used to determine if this leads to increased phagocytosis and increased survival when challenged with bacterial infection.

---

## [Author Response]

Essential revisions:

*The work done in the caterpillar is well done and nicely controlled. There are similar results already published for other insect species (Kim GS, Nalini M, Kim Y, Lee DW, Arch Insect Biochem Physiol. 2009; Kim GS and Kim Y, J Insect Physiol. 2010) and these need to be acknowledged up front in the introductory section with a sentence justifying why this study is still relevant. The reviewers found that the butterfly experiments are more detailed than in the older literature and that the extension to Drosophila opens up new possibilities for extending this research. Unfortunately, it is the fruit fly results that are not substantial enough to warrant an eLife publication. If some more characterization is presented in terms of phenotype and mechanisms, then this will achieve that goal. The authors are the best judges of the experiments that will further consolidate the fruit fly story. The manuscript will benefit from such addition. Of course, if necessary changes will take longer than 2 months, the authors could resubmit an expanded manuscript. The following major questions are provided as examples of questions that need to be addressed. 1) Is the receptor 5-HT_1B_ required autonomously in hemocytes? Does hemocyte-autonomous RNAi of 5-HT_1B_ affect the phagocytosis response, CFU after infection, and overall survival after infection? The authors could use Hml∆-GAL4 and ideally two independent UAS-RNAi lines. For suitable readouts of CFU and phagocytosis see below.*

We thank the reviewers for raising this point. As the reviewers suggested, we have used *Hml∆-GAL4* and two independent *UAS-5-HT_1B_* RNAi lines to address this question. As show in Figure 7, hemocyte-specific knockdown of 5-HT_1B_ decreased hemocyte phagocytic capacity, increased the bacterial load and susceptibility to *S. aureus*.

*2) Does 5-HT_1B_ indeed mediate phagocytic behavior in hemocytes and bacterial clearance as measured by CFU (or other hemocyte response/s)? Regarding the quantification of CFU after bacterial infection, it is unclear if bacterial strains with a resistance gene were used? This is not mentioned in the methods and needs to be clarified. Use of bacteria with a resistance marker is needed so injected bacteria can be distinguished from the naturally occurring bacterial load.*

*To carefully evaluate phagocytosis the authors should, at a minimum, use the trypan blue phagocytosis assay as described by Elrod-Erickson et al. Curr. Biol. 2000.*

We appreciate the reviewers’ constructive comments and suggestions. We are sorry for the unclear information regarding the bacterial strains we used. The bacterial strain *S. aureus* is resistant to tetracycline and the LB plates contain tetracycline. We have added the information in the Methods.

The bacteria we used to evaluate phagocytosis are *E. coli* and *S. aureus* labeled with pHrodo, a dye that fluoresces in the acidic environment of a mature phagosome upon fusion with lysosomes. This method reduces signal variability and improves assay reproducibility since wash steps and quencher dyes are not needed. It is now widely used in mammals (Fiala M et al., Proc Natl Acad Sci USA. 104:12849-12854, 2007; Berger SB et al., Nat Immunol.11:920-927,2010; Neaga A et al., J Immunol Methods. 2011) and insects including *Drosophila* (Cuttell L et al., Cell 135, 524-534, 2008; Ulvila J et al., J Leukoc Biol. 89(5): 649-59, 2011; Stone EFet al., PLoS Pathog. 8(1): e1002445, 2012; Lombardo Fet al., PLoS Pathog. 9(1): e1003145, 2013; Regan JC et al., PLoS Pathog. 9(10): e1003720, 2013; Garg A and Wu LP. Cell Microbiol. 16(2): 296-310, 2014). Besides, we also tested the classical trypan blue phagocytosis assay described by Elrod-Erickson et al. (Curr. Biol. 2000) to check the hemocyte phagocytic difference between *5-HT_1B_* null flies and control flies. The results are in Author response image 1 and similar to that by the pHrodo dye assays (Figure 6). Thus, we still used the pHrodo dye in phagocytosis assays of this paper.

Quantification of in vivo phagocytosis of *E. coli.*

*3) Does infection or LPS challenge indeed trigger an increased phagocytic hemocyte response through 5-HT_1B_?*

*Does hemocyte-specific kd of 5-HT_1B_ dampen phagocytosis in general, or is there indeed a lack of inducibility, e.g. after LPS challenge? Comparison of injecting fluorescent beads -/+LPS would allow to address this. A wt control should be used in parallel to confirm inducibility by the immune challenge here. The experiment could also be performed by injecting fluorescent beads with -/+ 5-HT.*

We thank the reviewers for this valuable comment. Following the reviewers’ suggestion, we used latex beads with -/+ 5-HT to address this question. Flies injected with PBS all show similar phagocytic capacity. However, when flies are injected with 5-HT, the flies expressing 5-HT_1B_ RNAi in blood cells take up significantly less latex beads than the control (Figure 7). These results indicate that hemocyte-specific knockdown of 5-HT_1B_ flies do not have a more limited phagocytic capacity, but instead become unable to enhance phagocytosis upon exposure to 5-HT.

*4) Do 5-HT levels increase upon bacterial infection in Drosophila? It would be easy to perform a hemolymph ELISA (as in Figure 1) to see if 5-HT levels increase upon infection in flies.*

We thank the reviewers for this suggestion. However, we have tried many times with different protocols but still failed to culture hemocytes in the medium without contamination for a longer time (>30 min). The small body size makes the *Drosophila* larvae extremely difficult to isolate clean hemolymph like we did on the caterpillar.

*5) Both 5-HT_1B_ and 5-HT_2B_ seem to be expressed in hemocytes of uninfected adult Drosophila (Figure 6). But the entire focus of the infection study phenotype is limited to the 1B mutant. What effects, if any, does inhibition/reduction of 2B have on phagocytosis and the lifespan of immune challenged animals? Would it be the same as in the caterpillar? If there is no mutant for 2B, the authors could minimally check levels of 2B expression in the hemocytes of infected animals, and perhaps one would see a down regulation similar to that in the caterpillar. There are MiMIC and TRiP lines available at BDSC, and minimally, 2B RNAi lines could be used to determine if this leads to increased phagocytosis and increased survival when challenged with bacterial infection.*

Following these comments, we have used *Hml∆-GAL4* and two independent *UAS-5-HT_2B_* RNAi lines to investigate the role of 5-HT_2B_ in phagocytosis and the lifespan of immune challenged animals. As Figure 8 shows, the reduced levels of 5-HT_2B_ in hemocyte led to the defect in phagocytosis of *E. coli* and *S. aureus* (Figure 8) and that was associated with higher bacterial load (Figure 8) and increased susceptibility to *S. aureus* (Figure 8) in the flies. The results suggest that activation of 5-HT_2B_ may promote phagocytosis in *Drosophila*, different from the results in the caterpillar.